# Improved outcomes over time for adult COVID-19 patients with acute respiratory distress syndrome or acute respiratory failure

Eric O. Yeates[1]☯*, Jeffry Nahmias[1‡], Justine Chinn[1‡], Brittany Sullivan[1], Stephen Stopenski[1], Alpesh N. Amin[2], Ninh T. Nguyen[1]☯

1 Department of Surgery, University of California Irvine, Orange, California, United States of America,
2 Department of Medicine, University of California Irvine, Orange, California, United States of America

☯ These authors contributed equally to this work.
‡ These authors also contributed equally to this work.
* yeatese@hs.uci.edu

## Abstract

### Background

COVID-19's pulmonary manifestations are broad, ranging from pneumonia with no supplemental oxygen requirements to acute respiratory distress syndrome (ARDS) with acute respiratory failure (ARF). In response, new oxygenation strategies and therapeutics have been developed, but their large-scale effects on outcomes in severe COVID-19 patients remain unknown. Therefore, we aimed to examine the trends in mortality, mechanical ventilation, and cost over the first six months of the pandemic for adult COVID-19 patients in the US who developed ARDS or ARF.

### Methods and findings

The Vizient Clinical Data Base, a national database comprised of administrative, clinical, and financial data from academic medical centers, was queried for patients $\geq$ 18-years-old with COVID-19 and either ARDS or ARF admitted between 3/2020-8/2020. Demographics, mechanical ventilation, length of stay, total cost, mortality, and discharge status were collected. Mann-Kendall tests were used to assess for significant monotonic trends in total cost, mechanical ventilation, and mortality over time. Chi-square tests were used to compare mortality rates between March-May and June-August.

110,223 adult patients with COVID-19 ARDS or ARF were identified. Mean length of stay was 12.1±13.3 days and mean total cost was $35,991±32,496. Mechanical ventilation rates were 34.1% and in-hospital mortality was 22.5%. Mean cost trended downward over time (p = 0.02) from $55,275 (March) to $18,211 (August). Mechanical ventilation rates trended down (p<0.01) from 53.8% (March) to 20.3% (August). Overall mortality rates also decreased (p<0.01) from 28.4% (March) to 13.7% (August). Mortality rates in mechanically ventilated patients were similar over time (p = 0.45), but mortality in patients not requiring mechanical ventilation decreased from March-May compared to June-July (13.5% vs 4.6%, p<0.01).

**Data Availability Statement:** All relevant data are within the manuscript and its S1 File.

**Funding:** The author(s) received no specific funding for this work.

**Competing interests:** I have read the journal's policy and the authors of this manuscript have the following competing interests: Alpesh Amin reported serving as PI or co-I of clinical trials sponsored by NIH/NIAID, NeuroRx Pharma, Pulmotect, Blade Therpeutics, Novartis, Takeda, Humanigen, Eli Lilly, PTC Therapeutics, OctaPharma, Fulcrum Therapeutics, Alexion. He has served as speaker and/or consultant for BMS, Pfizer, BI, Portola, Sunovion, Mylan, Salix, Alexion, AstraZeneca, Novartis, Nabriva, Paratek, Bayer, Tetraphase, Achogen LaJolla, Millenium, HeartRite, Aseptiscope, Sprightly. Ninh Nguyen reported serving as a speaker for Olympus and Endogastric Solutions. This does not alter our adherence to PLOS ONE policies on sharing data and materials.

## Conclusions

This study describes the outcomes of a large cohort with COVID-19 ARDS or ARF and the subsequent decrease in cost, mechanical ventilation, and mortality over the first 6 months of the pandemic in the US.

## Introduction

The severe acute respiratory syndrome coronavirus 2 (SARS-CoV-2), implicated in COVID-19, has been discovered to inflict an extremely wide spectrum of disease severity and manifestations [1, 2]. COVID-19's pulmonary manifestations are equally as broad, ranging from pneumonia with no supplemental oxygen requirements to acute respiratory distress syndrome (ARDS) with acute respiratory failure (ARF) [3]. In response to this novel spectrum of pulmonary disease, new oxygenation strategies and therapeutics have been developed at impressive speed. These include guidelines on prone positioning and high-flow nasal cannula, Remdesivir, corticosteroids, and convalescent plasma [4–9].

However, the large-scale effects of these advancements on outcomes in COVID-19 patients in the United States (US) remain unknown. Though recent studies have begun to describe these outcomes, they include patients spanning the full spectrum of COVID-19 disease, thereby evaluating a heterogeneous population [10, 11]. To our knowledge, no large study to date has described the changes in outcomes over time for COVID-19 patients with severe pulmonary disease.

Therefore, we aimed to examine the trends in mortality, mechanical ventilation, and cost over the first six months of the pandemic for adult COVID-19 patients in the US who developed ARDS or ARF. We hypothesized a downward trend in mortality, mechanical ventilation, and total cost over time.

## Materials and methods

The Vizient Clinical Data Base (VCDB), a US national database comprised of administrative, clinical, and financial data from academic and affiliated community medical centers, was queried for patients greater than 18 years old with an ICD-10 diagnosis of COVID-19 (UO7.1) and either ARDS (J80) or ARF (J960, J9600, J9601, J9602) admitted between March and August 2020. The Institutional Review Board of the University of California, Irvine deemed this study exempt from the need for consent as VCDB is deidentified.

The primary outcome was in-hospital mortality. Secondary outcomes included mechanical ventilation and total cost. Additional outcomes measured included length of stay and discharge status (including home, expired, skilled nursing facility, long-term care hospital, other facility, and hospice). Age, sex, race, and comorbidities (including hypertension, diabetes, obesity, congestive heart failure, renal failure, and anemia) were also collected. Categorical variables were expressed as numbers of patients with percentages and continuous variables were expressed as means with standard deviations.

Mann-Kendall tests were used to assess for significant monotonic trends in total cost, mechanical ventilation, and overall mortality rates over time. Mortality rates were further delineated into rates of those who were mechanically ventilated and those who were not. Mann-Kendall tests were again used to assess for trends within these two mortality rates. Additionally, Chi-square tests were used to compare mortality rates between March-May and June-

August. Finally, the percentage of COVID-19 positive patients out of all patients admitted each month was calculated to estimate the burden of disease on the hospital system over time. A Pearson correlation test was performed between these percentages and mortality rates over time. P-values less than 0.05 were considered statistically significant. All statistical analysis was performed using R 4.0.3 (R Core Team, 2020).

## Results

### Demographics and comorbidities

110,223 adult patients with COVID-19 and ARDS or ARF were identified. 42.8% were male, 43.3% were Caucasian, 26.2% were African American, and 3.9% were Asian. The most common age groups were 51–64 years old (29.8%) and greater than 75 years old (25.1%). The most common comorbidities were hypertension (66.0%), diabetes (42.4%), and obesity (32.9%) (Table 1).

### Outcomes

Mean length of stay (LOS) was 12.1 ± 13.3 days and mean total cost was $35,991 ± 32,496. Mechanical ventilation rates were 34.1% and in-hospital mortality was 22.5%. Mortality rates ranged from 5.6% for those 18–30 years old to 38.8% for those greater than 75 years old. 51.4% of patients were discharged to home, 20.4% to a skilled nursing facility/long-term care hospital/other facility, and 3.7% to hospice (Table 2).

**Table 1. Demographics and comorbidities of adults with COVID-19 and acute respiratory distress syndrome or acute respiratory failure.**

| Characteristic | N = 110,223 |
|---|---|
| Gender, No. (%) | |
| • Male | 63,003 (57.2) |
| • Female | 47,220 (42.8) |
| Race, No. (%) | |
| • Caucasian | 47,781 (43.3) |
| • African American | 28,851 (26.2) |
| • Asian | 4,332 (3.9) |
| • Other/unavailable/unknown | 29,259 (26.5) |
| Age, No. (%) | |
| • 18–30 years | 3,902 (3.5) |
| • 31–50 years | 22,112 (20.1) |
| • 51–64 years | 32,876 (29.8) |
| • 65–74 years | 23,646 (21.5) |
| • ≥ 75 years | 27,687 (25.1) |
| Comorbidities, No. (%) | |
| • Hypertension | 72,743 (66.0) |
| • Diabetes | 46,745 (42.4) |
| • Obesity | 36,228 (32.9) |
| • Congestive heart failure | 17,352 (15.7) |
| • Renal failure | 24,023 (21.8) |
| • Anemia | 26,180 (23.8) |

**Table 2. Outcomes of adults with COVID-19 and acute respiratory distress syndrome or acute respiratory failure.**

| Outcome | N = 110,223 |
|---|---|
| Mean length of stay (days) | 12.1 ± 13.3 |
| Mean total cost ($) | 35,991 ± 32,496 |
| Mechanical ventilation, No. (%) | 37570 (34.1) |
| In-hospital mortality, No. (%) | 24,799 (22.5) |
| In-hospital mortality according to age, No. (%) | |
| • 18–30 years | 220 of 3,902 (5.6) |
| • 31–50 years | 1,854 of 22,112 (8.4) |
| • 51–64 years | 5,435 of 32,876 (16.5) |
| • 65–74 years | 6,545 of 23,646 (27.7) |
| • ≥ 75 years | 10,745 of 27,687 (38.8) |
| Discharge status, No. (%) | |
| • Home | 56,613 (51.4) |
| • Expired | 24,799 (22.5) |
| • Skilled nursing facility/long-term care hospital/other facility | 22,463 (20.4) |
| • Hospice | 4,031 (3.7) |
| • Unknown and others[a] | 2317 (2.1) |

[a]Others: left against medical advice, transferred to other hospitals or healthcare institution not defined.

## Cost, mechanical ventilation, and mortality over time

Mean total cost trended downward over time (p = 0.02) from $55,275 in March to $18,211 in August. Mechanical ventilation rates trended down (p<0.01) from 53.8% in March to 20.3% in August. Overall mortality rates also decreased (p<0.01) from 28.4% in March to 13.7% in August (Table 3) (Figs 1 and 2).

## Mortality over time in mechanically ventilated and non-mechanically ventilated patients

Mortality rates in mechanically ventilated patients did not consistently downtrend over time (p = 0.45) (Table 3) (Fig 1). Mortality rates in mechanically ventilated patients were similar from March-May compared to June-July (46.8% vs 47.0%, p = 0.76).

Mortality rates in patients not requiring mechanical ventilation did not consistently downtrend over time (p = 0.13) (Table 3) (Fig 1). However, mortality rates in patients not requiring mechanical ventilation decreased from March-May compared to June-July (13.5% vs 4.6%, p<0.01).

## Proportion of COVID-19 positive admissions over time

The percentage of COVID-19 positive patients out of all patients admitted was 3.0% in March, 14.5% in April, 6.5% in May, 4.3% in June, 6.2% in July, 4.0% in August. A Pearson correlation test between these and mortality rates over the same months found no significant correlation (correlation coefficient = 0.43, p = 0.40).

## Discussion

Improved management strategies and novel therapeutics have been rapidly developed in the US for COVID-19 patients with ARDS or ARF, but nationwide changes in outcomes have not yet been described in this critically ill cohort. This large national study of adult patients with

**Table 3. Cost, mechanical ventilation, and mortality over time in adults with COVID-19 and acute respiratory distress syndrome or acute respiratory failure.**

| Outcomes | March (n = 12,417) | April (n = 39,742) | May (n = 16,966) | June (n = 12,232) | July (n = 19,679) | August (n = 8,726) | p-value |
|---|---|---|---|---|---|---|---|
| Mean total cost ($) | 55,275 | 36,607 | 37,874 | 35,161 | 28,690 | 18,211 | 0.02 |
| Mortality, No. (%) | 3,527 (28.4) | 10,912 (27.5) | 3,708 (21.9) | 2,062 (16.9) | 3,274 (16.6) | 1,194 (13.7) | <0.01 |
| Without mechanical ventilation, No. (%) | 5,738 (46.2) | 25,808 (64.9) | 11,127 (65.6) | 8,307 (67.9) | 14,423 (73.3) | 6,951 (79.7) | <0.01 |
| • Mortality, No. (%) | 693 (12.1) | 3,975 (15.4) | 1,091 (9.8) | 385 (4.6) | 648 (4.5) | 347 (5.0) | 0.13 |
| Mechanical ventilation, No. (%) | 6,679 (53.8) | 13,934 (35.1) | 5,839 (34.4) | 3,925 (32.1) | 5,256 (26.7) | 1,775 (20.3) | <0.01 |
| • Mortality, No. (%) | 2,834 (42.4) | 6,937 (49.8) | 2,617 (44.8) | 1,677 (42.7) | 2,626 (50.0) | 847 (47.7) | 0.45 |

COVID-19 and ARDS or ARF found a significant decrease in rates of mechanical ventilation, mortality, and cost over time. More specifically, we identified a decrease in mortality for patients not receiving mechanical ventilation.

A large percentage of critically ill COVID-19 patients required mechanical ventilation early in the pandemic, but a number of novel therapeutics and early oxygenation strategies (i.e. high flow nasal cannula treatment and early prone positioning) provided an impetus to reduce this need over the subsequent months of the pandemic [12, 13]. This study demonstrated that rates of mechanical ventilation in ARDS or ARF patients trended downward from March to August 2020 in the US. This may in part be due to novel therapeutics, including Remedesivir and corticosteroids, which may decrease the need for intubation [5–8]. Additionally, we suspect that prone positioning, high-flow nasal cannula, and a change in intubation triggers have also contributed to this noticeable reduction in mechanical ventilation rates [4, 14, 15].

We similarly identified a decrease in mortality over time. Though improved mortality compared to the start of the pandemic has been noted in previous studies, we now confirm this in a large, critically ill COVID-19 cohort in the United States [10, 11]. Furthermore, this study

**Fig 1. Mortality rates over time in adults with COVID-19 and acute respiratory distress syndrome or acute respiratory failure.** Mortality rates over time from March 2020 to August 2020 by mechanical ventilation status. "Overall" includes all patients, "Intubated" includes only patients that required mechanical ventilation, and "Not intubated" includes only patients that did not require mechanical ventilation during their hospital stay.

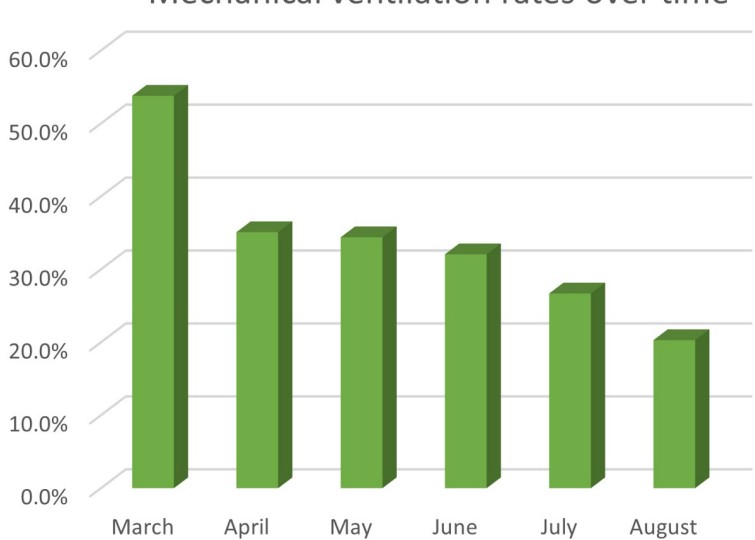

**Fig 2. Mechanical ventilation rates over time in adults with COVID-19 and acute respiratory distress syndrome or acute respiratory failure.** Mechanical ventilation rates over time from March 2020 to August 2020.

demonstrates that the change in mortality appears to originate from patients who were not mechanically ventilated, as the mortality in mechanically ventilated patients remained stable throughout the study period. This improvement is likely multifactorial but may be related to the novel use of corticosteroids and the increased adoption of non-invasive supplemental treatments including early prone positioning and high-flow nasal cannula [5–7]. These findings seem to also cast doubt that we have substantially improved mechanical ventilation strategies or made significant progress in treating the most severe cases of COVID-19. Another interesting finding was that the degree of burden of COVID-19 positive patients on the hospital system was not correlated with mortality. This suggests that the additional stress on the hospital system may not have worsened patient outcomes, however, we were unable to control for a number of confounding factors (i.e. novel treatments for COVID-19).

The COVID-19 pandemic could result in over 150 billion dollars in direct medical costs in the US due to the large number of infections and their poor outcomes [16, 17]. However, this study identified a decrease in mean total cost over the first 6 months of the pandemic in adult patients with COVID-19 and ARDS or ARF. This likely has a strong relationship to the decreased use of mechanical ventilation but may also be related to decreased length of stay and more efficient use of hospital resources [18]. Regardless, it is encouraging that, along with a significant improvement in patient outcomes, there has also been a reduction in the cost of care and burden on the US medical system throughout the pandemic.

This study has a number of limitations. Firstly, patients with ARDS or ARF were identified were using ICD-10 codes and were therefore subject to the discretion of many different clinicians. As the definition of ARDS in the context of COVID-19 was not initially widely agreed upon, it is possible that we were missing patients of interest in this study [3]. Next, we did not have access to patient-level data which made us unable to control for confounders (i.e. laboratory values, imaging findings, and baseline functional status) that may have been the true cause for the trends (or lack of trends) we identified in this study. Our lack of patient-level data also prevented us from attributing the improved outcomes (i.e. mortality, mechanical ventilation rates, and cost) to any specific intervention (i.e. Remedesivir, corticosteroids, and

prone positioning). Finally, it is also possible the patients most susceptible to severe COVID-19 died early during the pandemic and were therefore overrepresented in the earlier months, artificially improving outcomes in the later months [19, 20].

Despite these limitations, this study describes the outcomes of a large cohort with COVID-19 ARDS or ARF and the subsequent decrease in cost, mechanical ventilation, and mortality over the first 6 months of the pandemic in the US. This highlights the importance of stalling rapid spread of a pandemic to allow improved treatment and outcomes.

## Supporting information

**S1 File. Raw data for mechanical ventilation and mortality over time in adults with COVID-19 and acute respiratory distress syndrome or acute respiratory failure.** (XLSX)

## Author Contributions

**Conceptualization:** Eric O. Yeates, Jeffry Nahmias, Brittany Sullivan, Stephen Stopenski, Alpesh N. Amin, Ninh T. Nguyen.

**Data curation:** Eric O. Yeates, Justine Chinn, Ninh T. Nguyen.

**Formal analysis:** Justine Chinn, Ninh T. Nguyen.

**Methodology:** Eric O. Yeates, Justine Chinn, Alpesh N. Amin, Ninh T. Nguyen.

**Resources:** Eric O. Yeates.

**Supervision:** Jeffry Nahmias, Alpesh N. Amin, Ninh T. Nguyen.

**Writing – original draft:** Eric O. Yeates.

**Writing – review & editing:** Eric O. Yeates, Jeffry Nahmias, Justine Chinn, Brittany Sullivan, Stephen Stopenski, Alpesh N. Amin, Ninh T. Nguyen.

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
