## [Decision Letter · Decision Letter 0]

31 May 2021

PONE-D-21-09664

Improved outcomes over time for adult COVID-19 patients with acute respiratory distress syndrome or acute respiratory failure

PLOS ONE

Dear Dr. Yeates,

Thank you for submitting your manuscript to PLOS ONE. After careful consideration, we feel that it has merit but does not fully meet PLOS ONE’s publication criteria as it currently stands. Therefore, we invite you to submit a revised version of the manuscript that addresses the points raised during the review process.

Please revise accordingly.

We look forward to receiving your revised manuscript.

Kind regards,

Academic Editor

PLOS ONE

Journal Requirements:

[I have read the journal's policy and the authors of this manuscript have the following competing interests:

Alpesh Amin reported serving as PI or co-I of clinical trials sponsored by NIH/NIAID, NeuroRx Pharma, Pulmotect, Blade Therpeutics, Novartis, Takeda, Humanigen, Eli Lilly, PTC Therapeutics, OctaPharma, Fulcrum Therapeutics, Alexion. He has served as speaker and/or consultant for BMS, Pfizer, BI, Portola, Sunovion, Mylan, Salix, Alexion, AstraZeneca, Novartis, Nabriva, Paratek, Bayer, Tetraphase, Achogen LaJolla, Millenium, HeartRite, Aseptiscope, Sprightly.

Ninh Nguyen reported serving as a speaker for Olympus and Endogastric Solutions.].

Reviewers' comments:

Reviewer's Responses to Questions

**Comments to the Author**

1. Is the manuscript technically sound, and do the data support the conclusions?

Reviewer #1: Yes

Reviewer #2: Yes

2. Has the statistical analysis been performed appropriately and rigorously? 

Reviewer #1: Yes

Reviewer #2: Yes

3. Have the authors made all data underlying the findings in their manuscript fully available?

Reviewer #1: Yes

Reviewer #2: Yes

4. Is the manuscript presented in an intelligible fashion and written in standard English?

Reviewer #1: Yes

Reviewer #2: Yes

5. Review Comments to the Author

Reviewer #1: The authors describe outcomes over the first six months of the pandemic for adult COVID-19 patients in the US who developed acute respiratory distress syndrome or acute respiratory failure. This study describes the outcomes of a large cohort with COVID-19 ARDS or ARF and the subsequent decrease in cost, mechanical ventilation, and mortality over the first 6 months of the pandemic in the US. This is interesting and valuable to see the result in gradual reductions of mortality rates and costs with improvement treatment strategies and skills because no other research like this has been performed.

Are there any other factors that decreased the case mortality rate other than the improvement of medical technology? Reducing the burden on medical staff may be one of the factors that lowered the case mortality rate.

Is it possible to find out how many COVID-19 patients of all inpatients? If the authors knew the proportion of COVID-19 patients admitted to the facilities, the authors could infer how much burden has been placed on the medical facilities.

Reviewer #2: Eric O Yeates, et al. claims that mortality, cost, and the rate of mechanical ventilation of patients of COVID-19 with ARDS or ARF during March-May 2020 were significantly decreased compared with those during June-July 2020 in the USA. I am one of clinicians who have treated hundreds of hospitalized patients with COVID-19 in the country other than the US. The author’s statement is compatible with my clinical impression in my country. I agree with the most of their opinion. They focused on the patients with COVID-19 who experienced ARF or ARDS. Data focusing on the population are novel and interesting. However, there are a lot of limitations in the research, as they mention. Discussion mostly consisted of their assumption because they did not directly analyze the association between each factor (respiratory strategies, corticosteroid, or anti-viral medications) and the outcome. They only compared trends of March-May to those of June-July. It is the biggest limitation.

Major comments

1. As I pointed out, the author did not directly analyze the association between each factor and the outcome, such as mortality, ventilation-rate and the cost. It is the biggest limitation. So, they should mention as one of the study limitations.

2. Page 8 Line 131, I think that mortality rates in patients not requiring mechanical ventilation were similar over time, although the statistical significance was not shown. None-statistical significance dose not mean the similar. For example, I think that mortality in March (12.1%) is not similar to that in July (4.5%). Please revise the sentence.

3. Page 8 Line 147 to Line148, the authors should include corticosteroid into one of novel therapeutics. Corticosteroid is the most effective therapeutic agent against COVID-19. It has strong evidence.

Minor comments

1. Page 4 line 48, I think that the COVID-19 virus is correct medical jargon. SARS-CoV-2 is appropriate.

6. PLOS authors have the option to publish the peer review history of their article (what does this mean?). If published, this will include your full peer review and any attached files.

Reviewer #1: No

Reviewer #2: No

---

## [Author Response · Author response to Decision Letter 0]

10 Jun 2021

Reviewers' comments:

--------

Reviewer #1: The authors describe outcomes over the first six months of the pandemic for adult COVID-19 patients in the US who developed acute respiratory distress syndrome or acute respiratory failure. This study describes the outcomes of a large cohort with COVID-19 ARDS or ARF and the subsequent decrease in cost, mechanical ventilation, and mortality over the first 6 months of the pandemic in the US. This is interesting and valuable to see the result in gradual reductions of mortality rates and costs with improvement treatment strategies and skills because no other research like this has been performed.

Are there any other factors that decreased the case mortality rate other than the improvement of medical technology? Reducing the burden on medical staff may be one of the factors that lowered the case mortality rate.

Is it possible to find out how many COVID-19 patients of all inpatients? If the authors knew the proportion of COVID-19 patients admitted to the facilities, the authors could infer how much burden has been placed on the medical facilities.

Author response: Thank you for these insightful comments. The burden on medical staff and its effects on outcomes is something that has been commonly discussed, but not yet proven to our knowledge. To address this, we identified the percentage of COVID-19 positive patients out of all patients admitted each month and then performed a correlation test between those percentages and mortality. Interestingly, we did not find a correlation, suggesting that the additional stress on the hospital system may not have worsened patient outcomes. However, we admit there are a number of confounding factors not controlled for. We have now added the following interesting points into our Methods, Results, and Discussion:

“Finally, the percentage of COVID-19 positive patients out of all patients admitted each month was calculated to estimate the burden of disease on the hospital system over time. A Pearson correlation test was performed between these percentages and mortality rates over time.”

“The percentage of COVID-19 positive patients out of all patients admitted was 3.0% in March, 14.5% in April, 6.5% in May, 4.3% in June, 6.2% in July, 4.0% in August. A Pearson correlation test between these and mortality rates over the same months found no significant correlation (correlation coefficient=0.43, p=0.40).”

“Another interesting finding was that the degree of burden of COVID-19 positive patients on the hospital system was not correlated with mortality. This suggests that the additional stress on the hospital system may not have worsened patient outcomes, however, we were unable to control for a number of confounding factors (i.e. novel treatments for COVID-19).”

--------

Reviewer #2: Eric O Yeates, et al. claims that mortality, cost, and the rate of mechanical ventilation of patients of COVID-19 with ARDS or ARF during March-May 2020 were significantly decreased compared with those during June-July 2020 in the USA. I am one of clinicians who have treated hundreds of hospitalized patients with COVID-19 in the country other than the US. The author’s statement is compatible with my clinical impression in my country. I agree with the most of their opinion. They focused on the patients with COVID-19 who experienced ARF or ARDS. Data focusing on the population are novel and interesting. However, there are a lot of limitations in the research, as they mention. Discussion mostly consisted of their assumption because they did not directly analyze the association between each factor (respiratory strategies, corticosteroid, or anti-viral medications) and the outcome. They only compared trends of March-May to those of June-July. It is the biggest limitation.

Major comments

1. As I pointed out, the author did not directly analyze the association between each factor and the outcome, such as mortality, ventilation-rate and the cost. It is the biggest limitation. So, they should mention as one of the study limitations.

Author response: Thank you for this point. We agree this is one of the biggest limitations of the database we used and this study. We have now added the following to our Discussion to highlight this limitation:

“Our lack of patient-level data also prevented us from attributing the improved outcomes (i.e. mortality, mechanical ventilation rates, and cost) to any specific intervention (i.e. Remedesivir, corticosteroids, and prone positioning).”

2. Page 8 Line 131, I think that mortality rates in patients not requiring mechanical ventilation were similar over time, although the statistical significance was not shown. None-statistical significance dose not mean the similar. For example, I think that mortality in March (12.1%) is not similar to that in July (4.5%). Please revise the sentence.

Author response: Thank you for bringing this to our attention. We agree that this wording is misleading and have now changed this section of the Results to the following:

“Mortality rates in patients not requiring mechanical ventilation did not consistently downtrend over time.”

3. Page 8 Line 147 to Line148, the authors should include corticosteroid into one of novel therapeutics. Corticosteroid is the most effective therapeutic agent against COVID-19. It has strong evidence.

Author response: Thank you for pointing out this omission. We have now added corticosteroids to this portion of the Discussion and have also added two additional references on the topic.

“This may in part be due to novel therapeutics, including Remedesivir and corticosteroids, which may decrease the need for intubation [5-8].”

Minor comments

1. Page 4 line 48, I think that the COVID-19 virus is correct medical jargon. SARS-CoV-2 is appropriate.

Author response: Thank you. We have now changed this sentence to the following:

“The severe acute respiratory syndrome coronavirus 2 (SARS-CoV-2), implicated in COVID-19, has been discovered to inflict an extremely wide spectrum of disease severity and manifestations.”

---

## [Decision Letter · Decision Letter 1]

14 Jun 2021

Improved outcomes over time for adult COVID-19 patients with acute respiratory distress syndrome or acute respiratory failure

PONE-D-21-09664R1

Dear Dr. Yeates,

We’re pleased to inform you that your manuscript has been judged scientifically suitable for publication and will be formally accepted for publication once it meets all outstanding technical requirements.

Kind regards,

Academic Editor

PLOS ONE

Additional Editor Comments (optional):

Reviewers' comments:

Reviewer's Responses to Questions

**Comments to the Author**

1. If the authors have adequately addressed your comments raised in a previous round of review and you feel that this manuscript is now acceptable for publication, you may indicate that here to bypass the “Comments to the Author” section, enter your conflict of interest statement in the “Confidential to Editor” section, and submit your "Accept" recommendation.

Reviewer #1: All comments have been addressed

Reviewer #2: All comments have been addressed

2. Is the manuscript technically sound, and do the data support the conclusions?

Reviewer #1: Yes

Reviewer #2: Yes

3. Has the statistical analysis been performed appropriately and rigorously? 

Reviewer #1: Yes

Reviewer #2: Yes

4. Have the authors made all data underlying the findings in their manuscript fully available?

Reviewer #1: Yes

Reviewer #2: Yes

5. Is the manuscript presented in an intelligible fashion and written in standard English?

Reviewer #1: Yes

Reviewer #2: Yes

6. Review Comments to the Author

Reviewer #1: (No Response)

Reviewer #2: (No Response)

7. PLOS authors have the option to publish the peer review history of their article (what does this mean?). If published, this will include your full peer review and any attached files.

Reviewer #1: No

Reviewer #2: No

---

## [Editor Report · Acceptance letter]

18 Jun 2021

PONE-D-21-09664R1 

Improved outcomes over time for adult COVID-19 patients with acute respiratory distress syndrome or acute respiratory failure 

Dear Dr. Yeates:

I'm pleased to inform you that your manuscript has been deemed suitable for publication in PLOS ONE. Congratulations! Your manuscript is now with our production department. 

Kind regards, 

on behalf of

Dr. Robert Jeenchen Chen 

Academic Editor

PLOS ONE